# Effect of Proton Irradiation on Complementary Metal Oxide Semiconductor (CMOS) Single-Photon Avalanche Diodes

**Mingzhu Xun** [1,2,3], **Yudong Li** [1,2,*], **Jie Feng** [1,2], **Chengfa He** [1,2], **Mingyu Liu** [1,2,3] and **Qi Guo** [1,2]

1   Xinjiang Technical Institute of Physics and Chemistry, Chinese Academy of Sciences, Urumqi 830011, China; xunmz@ms.xjb.ac.cn (M.X.); fengjie@ms.xjb.ac.cn (J.F.); hecf@ms.xjb.ac.cn (C.H.); liumingyu21@mails.ucas.ac.cn (M.L.); guoqi@ms.xjb.ac.cn (Q.G.)
2   Xinjiang Key Laboratory of Electronic Information Material and Device, Urumqi 830011, China
3   University of Chinese Academy of Sciences, Beijing 100049, China
*   Correspondence: lydong@ms.xjb.ac.cn

**Abstract:** The effects of proton irradiation on CMOS Single-Photon Avalanche Diodes (SPADs) are investigated in this article. The I–V characteristics, dark count rate (DCR), and photon detection probability (PDP) of the CMOS SPADs were measured under 30 MeV and 52 MeV proton irradiations. Two types of SPAD, with and without shallow trench isolation (STI), were designed. According to the experimental results, the leakage current, breakdown voltage, and PDP did not change after irradiation at a DDD of $2.82 \times 10^8$ MeV/g, but the DCR increased significantly at five different higher voltages. The DCR increased by 506 cps at an excess voltage of 2 V and 10,846 cps at 10 V after 30 MeV proton irradiation. A $\gamma$ irradiation was conducted with a TID of 10 krad (Si). The DCR after the $\gamma$ irradiation increased from 256 cps to 336 cps at an excess voltage of 10 V. The comparison of the DCR after proton and $\gamma$-ray irradiation with two structures of SPAD indicates that the major increase in the DCR was due to the depletion region defects caused by proton displacement damage rather than the Si-SiO$_2$ interface trap generated by ionization.

**Keywords:** CMOS SPAD; proton radiation; DCR; displacement damage





## 1. Introduction

A Single-Photon Avalanche Diode (SPAD) is a photodiode that operates in Geiger mode with a reverse bias voltage higher than its avalanche breakdown voltage, and it utilizes an avalanche process to achieve single-photon detection capability. When photons are absorbed in the multiplication region of SPADs, a self-sustaining avalanche may be generated, and the current increases rapidly in the realm of picoseconds. By measuring the detectable current during the avalanche process, the arrival time of photons can be recorded [1]. An external quenching circuit is used to restore a SPAD to its initial state while waiting for the next photon to enter the multiplication region. By repeating the process above, single-photon detection and counting can be achieved. It can be seen that there is a dead time after the avalanche process, which may limit the maximum counting rate of the detected photons.

Due to their performance of high sensitivity, high detection efficiency, reliability, and low jitter noise, SPADs are widely used in some applications that require a low dark count rate (DCR), a low breakdown voltage, a low leakage current, a high gain, and high photon detection efficiency. SPADs are used in various fields, such as light detection and ranging (LiDAR), Non-Line of Sight (NLOS) imaging, fluorescence spectroscopy analysis, astronomic observations, optical communication, and quantum key distribution in weak light detection. When non-visible light must be used, especially in the near-infrared spectrum, high efficiency is very important [2].

At present, many research institutions and companies are developing SPADs with high efficiency, low noise, and high gain for different application scenarios, such as visible

and infrared light. In 2019, NASA reported an avalanche photodiode (APD) focal plane array assembled with linear-mode photon-counting capability for space lidar applications. The APD array uses a high-density, vertically integrated photodiode frame structure, and a preamplifier in the ROIC is directly integrated under the APD array to reduce the transmission capacitance. A microlens array is used to improve the fill factor. Its spectral response ranges from 0.9- to 4.3-μm wavelengths, its photon detection efficiency is as high as 70%, and it has a dark count rate of <250 kHz at 110 K [3]. In 2022, silicon photomultipliers (SiPMs), which are SPAD arrays based on a standard 55 nm Bipolar–CMOS–DMOS (BCD) technology, were developed by the Ecole Polytechnique Federale de Lausanne (EPFL). SiPMs are integrated into a coaxial light detection and ranging (LiDAR) system with a time-correlated single-photon counting (TCSPC) module system. Each SPAD cell is passively quenched by a monolithically integrated 3.3 V-thick oxide transistor. The measured gain is $3.4 \times 10^5$ at a 5 V excess bias voltage. The single-photon timing resolution (SPTR) is 185 ps, and the multiple-photon timing resolution (MPTR) is 120 ps at a 3.3 V excess bias voltage. Under the condition of a 25 m distance, the accuracies of SPTR and MPTR are 2 cm and 2 mm [4]. In 2022, a best-performing CMOS SPAD with a peak photon detection probability (PDP) of 55% at 480 nm, spanning from the near ultraviolet (NUV) to near infrared (NIR) spectrum, and a normalized dark count rate (DCR) of 0.2 cps/μm$^2$ at an excess bias of 6 V was proposed. Its after-pulsing probability is about 0.1% at a dead time of ∼3 ns, and its single-photon time resolution (SPTR) is 12.1 ps (FWHM) at a 6 V excess bias voltage with a diameter of 25 μm. SPADs operate over a wide range of temperatures, from −65 °C to 40 °C, reaching a normalized DCR of 1.6 mcps/μm$^2$ at a 6 V excess bias voltage and −65 °C [5]. Some big companies, such as STMicroelectronics, Sony, and HAMAMATSU, have also developed a series of SPADs for different applications.

In the field of radiation detection, SPAD arrays combined with different types of scintillators, which can absorb energy from radiation, are mainly used in high-sensitivity gamma-ray detectors and medical PET imaging [6–9]. Scintillator detectors are used for real-time radiation dose rate detection above the environmental background. PET imaging uses radioactive isotope tracing methods to display its location and concentration. By detecting the gamma photons generated by an isotope, the emission position of the photons can be reconstructed, and changes in metabolic processes and other physiological activities can be visualized. In addition, a single SPAD can be used to detect low-energy electrons and X-rays. A SPAD collects electrons generated by incident electrons and X-rays in the multiplication region instead of the photons emitted by scintillators. This makes detection faster and more accurate.

With the dramatic increase in interest in satellite-to-ground quantum communication and space environment detection, SPADs, with the advantages of high efficiency, low power consumption, easy integration, and anti-magnetic field performance, are more and more widely used in space and high-energy radiation detection [10–13]. But they are inevitably exposed to radiation environments, which can affect the performance of SPADs. Most satellite-to-ground quantum communication satellites are in near-earth orbit at an orbital altitude of 500 km, and the space radiation environment includes electrons and protons in the Van Allen radiation belt and high-energy protons in the South Atlantic Anomaly (SAA) region [14–17]. The space radiation environment during deep space exploration is dominated by high-energy galactic cosmic rays, including most of the particles in the periodic table from Z = 1 to Z = 92, with energies ranging from 1 MeV/n to 1 TeV/n. SiPMs were used in space-borne scintillation detectors for many space missions. For example, they have been used for the gain control system on board the Hard X-ray Modulation Telescope (HXMT), a Chinese X-ray space observatory launched in June 2017 [18]. The experiment GMOD assembled an SiPMs array with a CeBr3 scintillator and an Application-Specific Integrated Circuit (ASIC) to detect cosmic gamma-ray phenomena such as Gamma-Ray Bursts (GRBs) in space carried by the Educational Irish Research Satellite 1 (EIRSAT-1). This is a 2U cube satellite deployed from the International Space Station, and it remained in orbit at an altitude of 405 km and a tilt of 51.6 degrees for a year, which is a safe space

environment to avoid serious damage to SiPMs. SiPMs are also used in the Large Hadron Collider CMS, LHCb, and the proposed International Linear Collider (ILC) at the European Organization for Nuclear Research (CERN), which reaches $10^{14}$ p/cm$^2$ [19,20].

In radiation environments, protons, electrons, $\gamma$ rays, and heavy ions can cause certain parameters, such as the breakdown voltage, leakage current, DCR, gain, and photon detection efficiency, to deteriorate at different levels through the displacement damage dose (DDD) effect and the Total Ionizing Dose (TID) effect [21,22]. This is due to the point defects in silicon and the interface defects at the Si-SiO$_2$ interface near STI. These defects include the vacancy ($V_{Si}$), the substitutional phosphorus ($P_{Si}$), the interstitial oxygen (Oi), the double vacancy ($V_{Si}V_{Si}$), the A-center ($V_{Si}O_i$), and the E-center ($P_{Si}V_{Si}$). They are electrically active and act as efficient generation–recombination centers which cause leakage currents and DCR increases [23–27].

To study the SPAD radiation effect of protons, a SPAD of 180 nm standard CMOS technology with a P-I-N structure and radiation tolerance design is used in this experiment. The sensitivity of ionization radiation damage and displacement radiation damage for SPADs is investigated using $\gamma$ rays and protons beams. Two types of SPADs, with and without Shallow Trench Isolation (STI), are also designed and compared to study the influence of the Si-SiO$_2$ interface defects near the STI after radiation. The dark current, breakdown voltage, DCR, and photon detection probability (PDP) of the SPADs before and after irradiation are measured, and the radiation damage mechanism of the CMOS SPAD is analyzed.

## 2. Experimental Design

The CMOS SPAD, as shown in Figure 1, is based on the P-I-N structure, with a P-well epitaxial layer and n-type buried channel, and designed by the 180 nm CMOS process. Figure 2 is the cross-section of the SPAD [28,29]. In this design, the n-type buried channel ensures isolation from the substrate, while the deep n-well structure provides contact from N+ to the n-type buried channel. The lateral diffusion and light doping of the P epitaxial layer can avoid premature breakdown at the edge of the junction depletion region. The p+/DNW junction, enabling wider depletion, along with novel guard ring designs, facilitate device operation at up to 10 V of excess bias. The DCR is mainly caused by the tunneling noise at an excess bias of 10 V, but in this design, a P-I-N structure with standard CMOS technology is used to reduce the tunneling noise, resulting in better noise performance. The CMOS SPAD had a photon detection probability (PDP) greater than 40% from 440 to 620 nm, and the dark count rate (DCR) was 12.85 cps/μm$^2$. In addition, due to the use of n-type buried channels, the peak electric field of the detector is concentrated between the n-type buried channels and the P epitaxial layer.

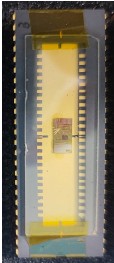

**Figure 1.** SPAD for 180 nm CMOS technology.

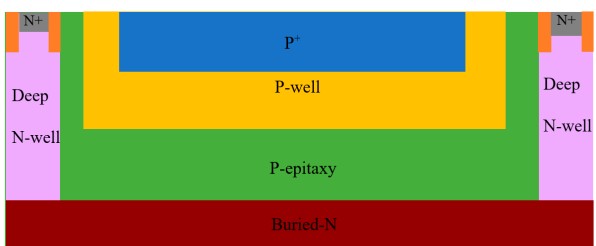

**Figure 2.** Cross-section of the P-I-N-structure SPAD.

Proton irradiation experiments at two different energies of 30 MeV and 52 MeV were conducted on the Cyclotron Proton Accelerator in the air, and a laser pointing system was used to align it with the beamline center. The proton beam region is 5 cm × 5 cm, and the uncertainty of the beam intensity had a variation of ±5%. The proton line energy transport (LET), Nonionizing Energy Loss (NIEL), Total Ionizing Dose (TID), and displacement damage dose (DDD) are shown in Table 1. The LET data come from the NIST stopping power and range tables for the protons program PSTAR, and the NIEL data come from Ref. [30]. The DDD of the SPAD in LEO orbit with an altitude of 400 km and an inclination of 51.6° was 19.6 TeV/g with a 2 mm shielding thickness of aluminum [31]. As a contrast, we chose a proton fluence of $5 \times 10^{10}$ p/cm$^2$ at energies of 30 MeV and 52 MeV. All the SPAD pins were shorted and connected to ground during the irradiation, and I–V characteristics and PDP measurements were performed before and after irradiation. The parameter testing system for a SPAD includes a Keysight semiconductor parameter analyzer and a DCR and PDP measurement system. The DCR and PDP measurement system consists of a light source, a filter, a spectrograph, an integrating sphere, a sample chamber, and a light source calibration and computer control system, as shown in Figure 3. The halogen lamps can provide a stable light source with a wavelength range of 350 nm–1100 nm. Monochromatic light with specific frequencies can be generated after light passes through the filters and spectrometers. The integrating sphere can reduce small errors caused by an uneven distribution of incident light sources on the detector or beam offset during measurement, thus improving the accuracy of a measurement. The output trigger pulse count of the device is read by an oscilloscope, and then the data are statistically analyzed to obtain the DCR and PDP. A passive quenching circuit with a 50 kΩ resistor was used to measure the DCR, and the DCR is defined as the average counts of pulses per second (cps) in a 1 min measurement in darkness.

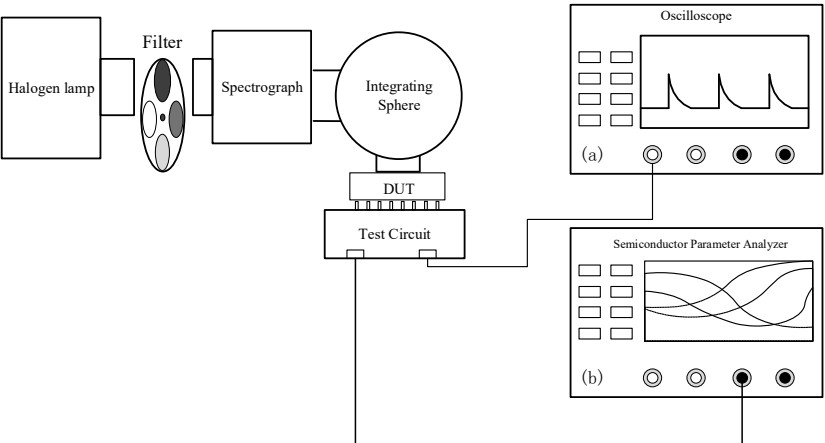

**Figure 3.** The CMOS SPAD parameter testing system with (**a**) oscilloscope for DCR and PDP measurement and (**b**) semiconductor parameter analyzer for I–V characteristics measurement.

**Table 1.** LET, NIEL, TID, and DDD of 30 MeV and 52 MeV protons.

| Proton Energy (MeV) | LET (MeV/(g/cm$^2$)) | NIEL (MeV/(g/cm$^2$)) | Fluence (p/cm$^2$) | TID (krad) | DDD (MeV/g) |
|---|---|---|---|---|---|
| 30 | $1.47 \times 10$ | $5.63 \times 10^{-3}$ | $5.00 \times 10^{10}$ | 11.8 | $2.82 \times 10^8$ |
| 52 | $9.58 \times 10$ | $3.37 \times 10^{-3}$ | $5.00 \times 10^{10}$ | 7.66 | $1.69 \times 10^8$ |

## 3. Results

### 3.1. I–V Characteristics

Figure 4 shows the I–V characteristics of the CMOS SPAD. The leakage current of the SPAD after 30 MeV proton irradiation did not increase significantly before reaching the avalanche breakdown voltage. When a bias voltage of 26 V was applied, the reverse current increased from 5.47 mA to 5.71 mA. Table 2 shows a comparison of the SPAD breakdown voltages. The breakdown voltage increased by only 20 mV after 30 MeV proton irradiation, while it remained unchanged after 52 MeV proton irradiation. This indicates that proton displacement damage can lead to a slight increase in the breakdown voltage, but not significantly. This is similar to the results of SPAD $\gamma$ experiments based on the same P-I-N structure in Ref. [28], indicating that the leakage current and breakdown voltage ($V_B$) are almost insensitive to ionization damage and displacement damage.

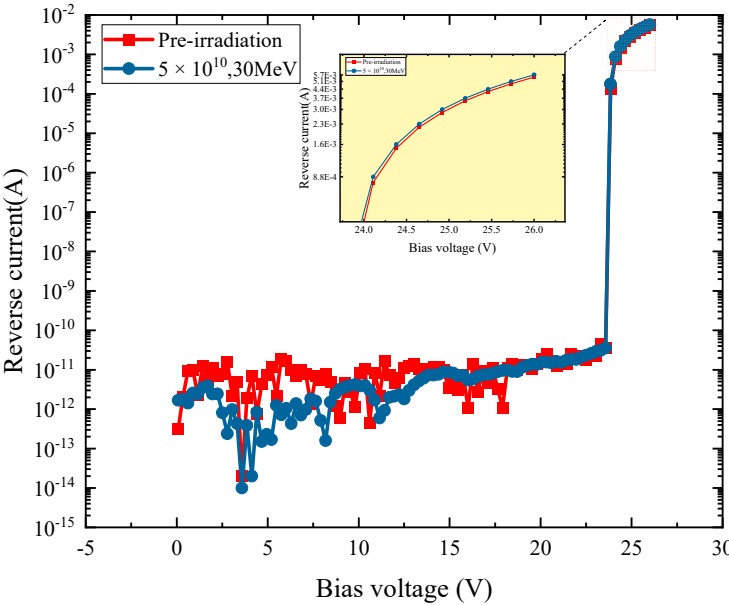

**Figure 4.** I–V characteristics for CMOS SPAD.

**Table 2.** Comparison of the SPAD breakdown voltage.

| Proton Energy (MeV) | $V_B$ (Fresh) (V) | $V_B$ ($5 \times 10^{10}$ p/cm$^2$) (V) |
|---|---|---|
| 30 | 24.23 | 24.25 |
| 52 | 24.18 | 24.18 |

### 3.2. DCR

Figure 5 shows the DCR data under different excess voltages before and after irradiation with a passive quenching circuit at 23 °C. It can be seen that before irradiation, the DCR increases with the increase in excess voltage from 74cps@2V to 256cps @10V. But after proton irradiation, the increase in the DCR is very significant. The DCR increases from 74cps@2V before irradiation to 520cps@2V after 52 MeV proton irradiation. This is consistent with the trend of DCR change with excess voltage before irradiation. However,

due to the increase in displacement damage defects in the junction depletion region caused by proton irradiation, the trend of DCR increase with higher excess voltage is significantly enhanced. The DCR increased form 580cps@2V to 11102cps@10V after 30 MeV proton irradiation. Under the same proton fluence of $5 \times 10^{10}$ p/cm$^2$, the change caused by 30 MeV proton irradiation is greater than that caused by 52 MeV proton irradiation. This is due to the NIEL of low-energy protons being higher than that of high-energy protons, resulting in a DDD of $2.82 \times 10^8$ MeV/g for the 30 MeV protons, which is higher than the DDD of $1.69 \times 10^8$ MeV/g for the 52 MeV protons. So, the displacement damage defects generated by the 30 MeV protons in the junction depletion region resulted in a larger DCR at the same excess voltage.

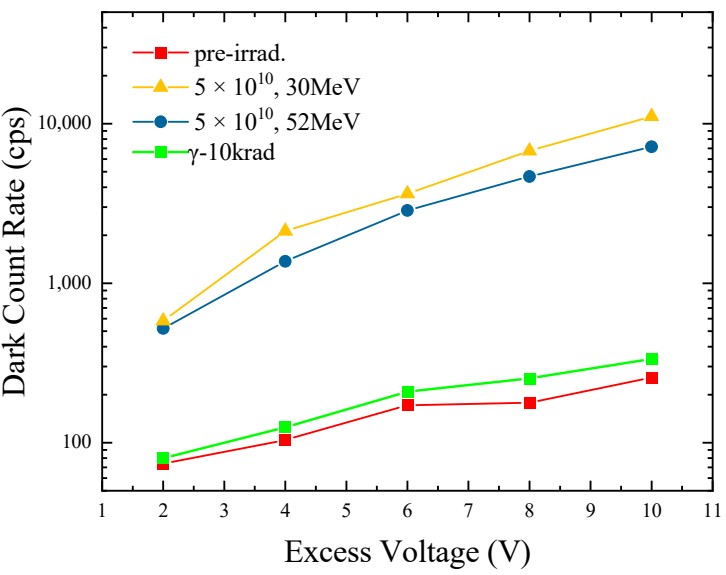

**Figure 5.** DCR of different excess voltages after proton and $\gamma$ irradiation.

The TID and DDD caused by proton irradiation resulted in Si-SiO$_2$ interface defects and junction depletion region defects. In order to confirm the main reason for the increase in the DCR, $\gamma$ irradiation was conducted with a TID of 10 krad(Si), while the TID of the 52 MeV proton was only 7.66 krad(Si). The DCR after $\gamma$ irradiation increased from 256 cps to 336 cps at an excess voltage of 10 V. However, after proton irradiation, the DCR increased to 7160 cps, which is approximately 20 times greater than that of $\gamma$ irradiation, indicating that the increase in the DCR is mainly caused by the displacement damage of proton irradiation, and the TID effect is not obvious.

### 3.3. PDP

The PDP is defined as the ratio of the SPAD-detected photons to the incident photons and reflects the generation of photo-generated carriers. It reflects the photosensitivity of the SPAD. The PDP depends on two main parameters: the absorption probability and the triggering efficiency. The absorption probability is the probability of photons being absorbed in the depletion region, and it depends on the reflectivity, the depth of the junction, and the thickness of the depletion region, while the triggering efficiency is the probability of photo-generated electron–hole pairs triggering a self-sustaining avalanche process, which depends on the electric field [1]. Figure 6 shows a comparison of the PDP curve at an excess voltage of 6 V before and after 30 MeV proton irradiation. It can be seen that there is no significant change in the PDP in the wavelength range of 400 nm–800 nm. The maximum PDP of the SPAD is 37.9% and 38.6% at a wavelength of 500 nm. This indicates that the depletion region defects caused by a proton displacement irradiation damage dose of $2.82 \times 10^8$ MeV/g have no effect on the absorption of incident photons and the generation of photo-generated carriers, so the PDP does not change. Ref. [28] reports that the PDP

also has no changes after $\gamma$ irradiation, which also means that the PDP is not an important radiation-sensitive parameter to consider in radiation environments [32].

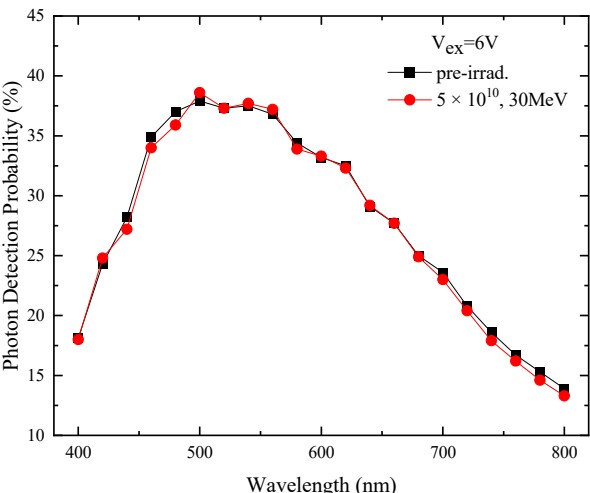

**Figure 6.** PDP value of 30 MeV protons.

## 4. Discussion

From the previous results, it can be concluded that the SPAD leakage current, breakdown voltage, and PDP are not sensitive to proton displacement damage, but the DCR is very sensitive. The DCR reflects the inherent noise inside a single-photon detector. The sources of noise in silicon devices include thermal noise, tunneling-assisted noise, and trap-assisted noise. Among them, thermal noise and tunneling-assisted noise are related to the operating temperature, doping concentration, and excess voltage of the device. Trap-assisted noise is related to defects introduced during the CMOS manufacturing process, and trap defects introduced during irradiation also produce trap-assisted noise [33–37]. The thermal generation and band-to-band tunneling effects of free carriers within the depletion region collectively contribute to the DCR, which is largely dependent on temperature. When the temperature increases by 10 °C near the room temperature of 23 °C, the DCR usually increases by more than double [38,39]. At this temperature, thermal generation is the main noise source. The thermal generation of free carriers is closely related to the presence of impurities and crystal defects, which introduce local energy levels near the middle of the band gap. According to the Shockley–Read–Hall theory, electron–hole pairs are generated sequentially through generation–recombination (G-R) centers. The proton irradiation of silicon-based devices can also form deep-energy-level defects near the center of the energy band in the depletion region, which can trap or emit electrons and become the trap center. These irradiated deep-energy-level defects contribute to the increase in the dark count rate. Besides the thermal effect, another major contributor to the DCR are Poole–Frenkel effects and rap-assisted tunneling, but the DCR does not significantly increase with temperature but instead increases with excess bias, which usually happens in high-doping junctions. The effects of displacement damage on semiconductor materials and devices can be understood in terms of the energy levels introduced in the bandgap. Those radiation-induced levels result in the following effects: the recombination lifetime and diffusion length are reduced; the generation lifetime decreases; majority-carrier and minority-carrier trapping increase; the majority-carrier concentration changes; the thermal generation of electron–hole pairs is enhanced in the presence of a sufficiently high electric field; tunneling at junctions is enabled; and radiation-induced defects reduce the carrier mobility and can exhibit metastable configurations [40,41].

To analyze the displacement damage of protons in the SPAD, we used The Stopping and Range of Ions in Matter Software (SRIM) to simulate the proton transportation process, which is a program written by J.F. Ziegler, M.D. Ziegler, and J.P. Biersack to simulate the

interaction process of ion beams with solids, and the Monte Carlo method was used to calculate details such as vacancies, energy deposition, and particle positions during the collision process [42].

The simulation results of the proton trajectory, stopping power, and energy deposition per unit distance in the photo collector region are shown in Figure 7. At the SPAD surface, the stopping power of silicon for the 30 MeV protons is higher than that for the 52 MeV protons, indicating that the damage caused by the 30 MeV protons on silicon is more severe under the fluence of $5 \times 10^{10}$ p/cm$^2$. As the incident depth increases, the deposited energy of the 30 MeV protons in silicon increases significantly, which is manifested as an enhancement of proton scattering in the same number of particle trajectories, resulting in a larger projected area of the incident direction. The calculated number of vacancies in silicon is shown in Table 3. Comparing the difference between the vacancy numbers and the ΔDCR under different excess voltages, it is found that the number of vacancies is 24 for each 30 MeV proton and 13.7 for each 52 MeV proton, and the ratio is 1.75. However, the ratio of DCR increase at the five excess voltages varies from 0.93 to 1.59, which is a little lower than the ratio of vacancy. This may be because not all trap vacancies contribute to the generation of carriers, which will cause the DCR in the depletion region to increase; some vacancies are used for the carrier's recombination. The DCR increase is defined as

$$\Delta DCR = DCR \text{ after irradiation} - DCR \text{ before irradiation} \qquad (1)$$

**Table 3.** The ratio of vacancies and ΔDCR at different excess voltages.

| Proton Energy (MeV) | Total Vacancies (/ion) | ΔDCR@2V (cps) | ΔDCR@4V (cps) | ΔDCR@6V (cps) | ΔDCR@8V (cps) | ΔDCR@10V (cps) |
|---|---|---|---|---|---|---|
| 30 | 24 | 506 | 2020 | 3468 | 6584 | 10,846 |
| 52 | 13.7 | 446 | 1268 | 2686 | 4488 | 6904 |
| ratio | 1.75 | 1.13 | 1.59 | 1.29 | 1.47 | 1.57 |

In order to analyze the influence of Si-SiO$_2$ interface defects on the STI structure, two types of SPAD units, with and without an STI structure, were designed on the same chip. The simulation results of the SPAD design and the electric field distribution with and without an STI structure are shown in Figure 8. Between the p-well and deep n-well, we designed an STI structure using silicon dioxide as an insulating layer. It can be seen that the electric field distribution near the STI structure changes significantly. After ionizing radiation, interface charges accumulate at the Si-SiO$_2$ interface near the STI structure.

After proton irradiation with the same fluence, the DCR was measured under different excess voltages. The results are shown in Figure 9. The comparison of the results with and without an STI structure before irradiation shows that the presence of the STI structure increases the DCR from 256 cps to 362 cps under a 10 V bias. The interface defects in the STI structure before irradiation increase the DCR by 41.4%. Ref. [28] has proven that the increase in the DCR caused by ionizing irradiation is mainly due to the induced Si-SiO$_2$ interface traps near the STI structure. The DCR of the SPAD with and without an STI structure significantly increased after irradiation. At 10 V, the DCR increased by 30 times. However, when the excess voltage was 0–8 V, the DCR of the SPAD without an STI structure was higher than that of the SPAD with an STI structure. When the excess voltage was greater than 8 V, the DCR of the SPAD without the STI structure was lower. A possible reason for this is that the Si-SiO$_2$ interface defect charges near the STI structure will be released from the interface and drift into the depletion region when the electric field exceeds a certain value, and then an avalanche process is formed, resulting in an increase in the DCR.

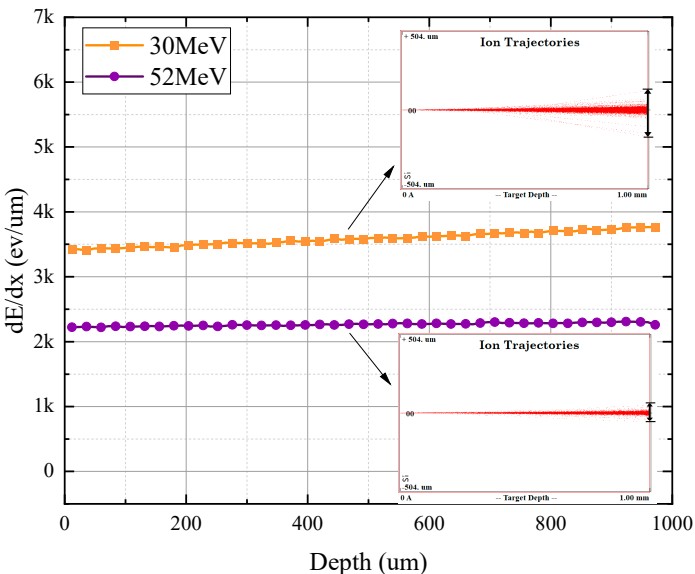

**Figure 7.** Stopping power and trajectory of protons in silicon.

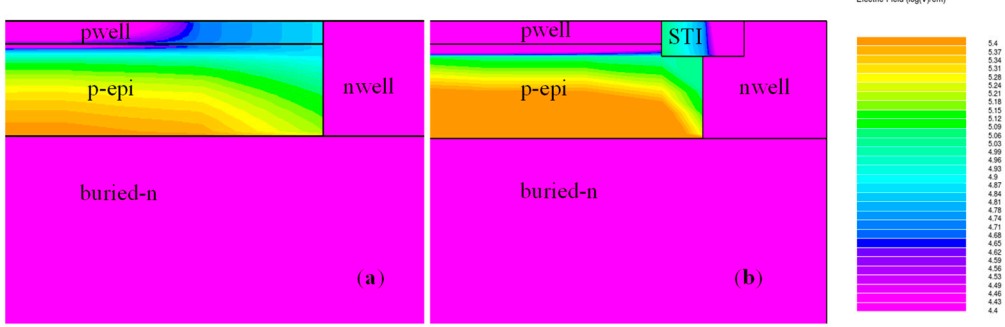

**Figure 8.** SPAD and electric field distribution without STI structure (**a**) and with STI structure (**b**).

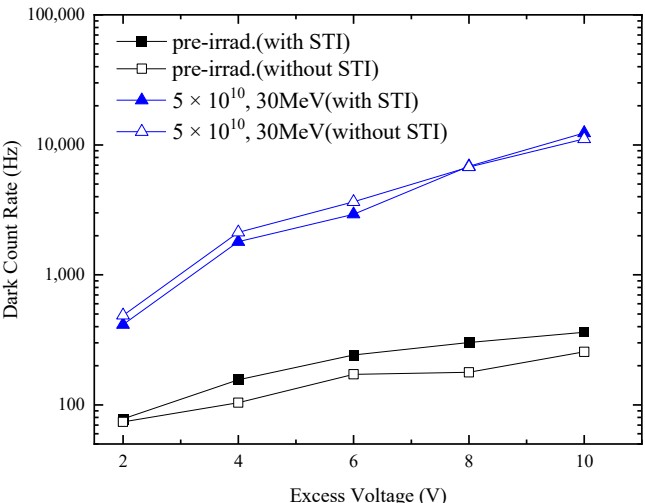

**Figure 9.** Comparison of DCR irradiation with and without STI structure.

## 5. Conclusions

Proton irradiations on the CMOS SPAD of 30 MeV and 52 MeV are studied in this article. The leakage current, breakdown voltage, and PDP before and after proton irradiations were measured and compared with γ rays. SPAD units with and without an STI structure were designed and simulated, and we summarize our results as follows:

1. After the 30 MeV proton radiation, the reverse current increased from 5.47 mA to 5.71 mA at a bias voltage of 26 V. The breakdown voltage increased by only 20 mV after the 30 MeV proton irradiation, while it remained unchanged after the 52 MeV proton irradiation. The reported results of the SPAD γ experiments based on the same P-I-N structure in Ref. [17] indicate that the breakdown voltage ($V_B$) is almost insensitive to ionization damage and displacement damage.

2. Before irradiation, the DCR increased with an increase in the excess voltage from 74cps@2V to 256cps@10V, but the DCR increased rapidly to 520cps@2V and 7160cps@10V after the 52 MeV proton irradiation. For the 30 MeV proton irradiation, the DCR increased form 580cps@2V to 11102cps@10V. The displacement damage defects generated by the 30 MeV protons with a DDD of $2.82 \times 10^8$ MeV/g resulted in a larger DCR increase than a DDD of $1.69 \times 10^8$ MeV/g for the 52 MeV protons at the same excess voltage. The trend of the DCR increasing with a higher excess voltage is significantly enhanced due to the displacement damage defects in the junction depletion region. A comparison of γ irradiation with a TID of 10 krad (Si) and the 52 MeV protons with a TID of 7.66 krad (Si) shows that the increase in the DCR is mainly caused by the displacement damage of proton irradiation instead of the TID effect.

3. The SPAD units with and without an STI structure also show that the main reason for the DCR increase is the depletion region defects caused by proton displacement damage rather than the Si-SiO$_2$ interface trap generated by ionization.

4. The comparison of the leakage current, breakdown voltage, and PDP shows that the design of the SPAD based on the standard CMOS process exhibits good radiation hardening, but the process of the depletion region should be improved to reduce the DCR after irradiation.

**Author Contributions:** Conceptualization, M.X. and Y.L.; methodology, M.X. and Y.L.; software, M.X.; validation, M.X. and Y.L.; formal analysis, Y.L.; investigation, Y.L.; resources, M.L. and C.H.; data curation, J.F.; writing—original draft preparation, M.X.; writing—review and editing, M.X. and Y.L.; visualization, M.X.; supervision, Y.L.; project administration, Q.G.; funding acquisition, Y.L. and Q.G. All authors have read and agreed to the published version of the manuscript.

**Funding:** This research was funded by the West Light Talent Training Plan of the Chinese Academy of Sciences under grant No. 2021-XBQNXZ-020, the Tianshan Innovation Team Program of Xinjiang Uygur Autonomous Region No. 2022D14003, the Fund of Robot Technology Used for Special Environment Key Laboratory of Sichuan Province No. 21kftk03, and the "Light of West China" Program of the Chinese Academy of Sciences under grant No. 2020-XBQNXZ-004.

**Data Availability Statement:** Data are contained within the article.

**Conflicts of Interest:** The authors declare no conflicts of interest.

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
