# Peer review of "Effect of Proton Irradiation on Complementary Metal Oxide Semiconductor (CMOS) Single-Photon Avalanche Diodes"

_electronics, doi:10.3390/electronics13010224_

Round 1
Reviewer 1 Report
Comments and Suggestions for Authors
1. What is the significance of the 20 mV increase in the breakdown voltage?
2. Figure 9, pre-irrad with and without STI are not able to be differentiated, and not able to view their trend.
3. Figure 8, please label the difference between (a), and (b).
4. What is the depth of the deep N-well?
Comments on the Quality of English Languageminor.
Author Response
- What is the significance of the 20 mV increase in the breakdown voltage?
Answer:On line 172,we added a sentence:“This indicates that proton displacement damage can lead to a slight increase in breakdown voltage, but not significantly.”
- Figure 9, pre-irrad with and without STI are not able to be differentiated, and not able to view their trend.
Answer:The Y-scale has been changed to log, and pre-irrad DCR can been seen clearly.
- Figure 8, please label the difference between (a), and (b).
Answer:Figure 8 has been modified and the difference has been labeled.
- What is the depth of the deep N-well?
Answer:The depth of the deep N-well is about 5um.

Reviewer 2 Report
Comments and Suggestions for Authors
The paper proposes an interesting study on Effect of Proton Irradiation on CMOS SPAD, but some aspects must be improved, in my opinion, before publication.
Row 69-79 You should add some references related to the use of SPAD in high-sensitivity gamma ray detectors and medical PET imaging
You should add some references related to the analysys of radiation damage on SPADs as:
- Wu, M.-L.; Ripiccini, E.; Kizilkan, E.; Gramuglia, F.; Keshavarzian, P.; Fenoglio, C.A.; Morimoto, K.; Charbon, E. Radiation Hardness Study of Single-Photon Avalanche Diode for Space and High Energy Physics Applications. Sensors 2022, 22, 2919. https://doi.org/10.3390/s22082919
- F. Moscatelli et al. “Radiation tests of single photon avalanche diode for space applications”, Nuclear Instruments and Methods in Physics Research A 711 (2013) 65–72
- M Campajola et al 2019 J. Phys.: Conf. Ser. 1226 012007
- Qiaoli Liu et al 2020 Chinese Phys. B 29 088501
Row 144 Why have you chosen a proton fluence of 5×1010 p/cm2? Have you considered lower or upper fluences? Explain why you have considered only this fluence in relation to the application (orbit, mission lifetime ….)
The devices are biased during the irradiation?
I suggest to modify Fig. 5 and Fig. 9 considering a log scale for the Y-axis.
Section 3.2 At which temperature have you measured the DCR? You write room temperature but can you measure the temperature? Have you measured the DCr at lower temperatures?
Have you analyzed the afterpulsing probability?
Have you seen ‘‘random telegraph signal’’ effect?
Have you considered annealing effects?
Author Response
The paper proposes an interesting study on Effect of Proton Irradiation on CMOS SPAD, but some aspects must be improved, in my opinion, before publication.
- Row 69-79 You should add some references related to the use of SPAD in high-sensitivity gamma ray detectors and medical PET imaging
Answer:Four references about the use of SPAD in high-sensitivity gamma ray detectors and medical PET imaging was added in the Row 72
- You should add some references related to the analysys of radiation damage on SPADs as:
- Wu, M.-L.; Ripiccini, E.; Kizilkan, E.; Gramuglia, F.; Keshavarzian, P.; Fenoglio, C.A.; Morimoto, K.; Charbon, E. Radiation Hardness Study of Single-Photon Avalanche Diode for Space and High Energy Physics Applications. Sensors 2022, 22, 2919. https://doi.org/10.3390/s22082919
- F. Moscatelli et al. “Radiation tests of single photon avalanche diode for space applications”, Nuclear Instruments and Methods in Physics Research A 711 (2013) 65–72
- M Campajola et al 2019 J. Phys.: Conf. Ser. 1226 012007
- Qiaoli Liu et al 2020 Chinese Phys. B 29 088501
Answer:The references has been added in the passage
- Row 144 Why have you chosen a proton fluence of 5×1010 p/cm2? Have you considered lower or upper fluences? Explain why you have considered only this fluence in relation to the application (orbit, mission lifetime ….)
Answer:In the row 144, we added the explanation “The DDD of SPAD in LEO orbit with the altitude of 400km and the inclination of 51.6° is 19.6TeV/g after the 2mm shielding thickness of aluminum[31]. As a contrast, we choose the proton fluence of 5E10 p/cm2 at the energy of 30MeV and 52MeV.”
- The devices are biased during the irradiation?
Answer:In the row 144, we added the sentence” All the SPAD pins are shorted and connect to ground during the irradiation,”
- I suggest to modify Fig. 5 and Fig. 9 considering a log scale for the Y-axis.
Answer:The Figure has been modified
- Section 3.2 At which temperature have you measured the DCR? You write room temperature but can you measure the temperature? Have you measured the DCr at lower temperatures?
Answer:We measured the DCR in a clean room in the lab, the stationary temperature is 23°C.In the passage, we modified the room temperature to 23°C. I didn’t measure the DCR at lower temperatures for this device.
- Have you analyzed the afterpulsing probability?
Answer:I didn’t analyze the afterpulsing probability for this device, but I will consider about it in the future.
- Have you seen ‘‘random telegraph signal’’ effect?
Answer:I didn’t seen ‘‘random telegraph signal’’ effect for this device because of the oscilloscope, a counter will be added to analyze RTS in the future.
- Have you considered annealing effects?
Answer:I didn’t consider the annealing effects for this device, but I will consider about it in the future.

Reviewer 3 Report
Comments and Suggestions for Authors
The present paper addresses the question of the effect of proton irradiation on SPADs, more precisely the effects on the I-V curve, the DCR, and the PDP. The paper is very clear and well written. The subject is very interesting and the conclusions very relevant. The paper can be published in Electronics but I ask to the Authors to correct the following points:
1. The introduction in a little bit too long. I would suppress the paragraph between lines 69 and 79.
2. The Authors have to define the acronyms NIEL, TID and DDD on line 142.
3. The last version of SRIM was explained by J.F. Ziegler, M.D. Ziegler and J.P. Biersack in Nuclear Instruments and Methods in Physics Research Section B, 2010, Volume 268, Issue 11-12, p. 1818-1823. The Authors have to add the other names on line 254 and to correct the reference.
Author Response
- The introduction in a little bit too long. I would suppress the paragraph between lines 69 and 79.
Answer: Thanks for your advice, but the usage of SPAD in the high-sensitivity gamma ray detectors and medical PET imaging also important for the introduction, four references about the use of SPAD in high-sensitivity gamma ray detectors and medical PET imaging was added in the Row 72
- The Authors have to define the acronyms NIEL, TID and DDD on line 142.
Answer: the acronyms have been defined
- The last version of SRIM was explained by J.F. Ziegler, M.D. Ziegler and J.P. Biersack in Nuclear Instruments and Methods in Physics Research Section B, 2010, Volume 268, Issue 11-12, p. 1818-1823. The Authors have to add the other names on line 254 and to correct the reference.
Answer: The other names and the correct reference has been added

Round 2
Reviewer 2 Report
Comments and Suggestions for Authors
I have no further comments